# Immunoglobulin Gene Sequence as an Inherited and Acquired Risk Factor for Chronic Lymphocytic Leukemia

**DOI:** 10.3390/cancers14133045

**Published:** 2022-06-21

**Authors:** Moumita Datta, Hassan Jumaa

**Affiliations:** Institute of Immunology, University Medical Center, 89081 Ulm, Germany

**Keywords:** chronic lymphocytic leukemia, immunoglobulin, autonomous signaling

## Abstract

**Simple Summary:**

Chronic lymphocytic leukemia (CLL) is the most prevalent among adult leukemias. Over the years, several research efforts discovered a lot of intricate details about the cause of the disease, its mechanism, and the prognostic factors that help to understand the progression and outcome of the disease. Mutations in the immunoglobulin gene sequences in B cells are the most important prognostic factor for CLL. The cells having no to very less mutations show aggressive disease, while those having more mutations are either fairly indolent or non-aggressive. In this review, we discussed the current gain of knowledge about these mutations and their effects in the overall disease pathology.

**Abstract:**

Chronic lymphocytic leukemia (CLL) is a lymphoproliferative disease characterized by the accumulation of CD5^+^ CD19^+^ malignant B cells. Autonomous ligand-independent B-cell signaling is a key process involved in the development of CLL pathogenesis. Together with other cytogenetic alterations, mutations in the immunoglobulin heavy chain variable (IGHV) gene act as a prognostic marker for CLL, with mutated CLL (M-CLL) being far more indolent than unmutated CLL (U-CLL). Recent studies highlight the role of a specific light chain mutation, namely, IGLV3-21^R110G^, in the development and prognosis of CLL. Such a mutation increases the propensity of homotypic BCR–BCR interaction, leading to cell autonomous signaling. In this article, we review the current findings on immunoglobulin gene sequence mutations as a potential risk factor for developing CLL.

## 1. Introduction

### 1.1. CLL Overview

Chronic lymphocytic leukemia (CLL) is a B-cell malignancy that has maximum incidence among all types of adult leukemia [1,2,3,4]. On the molecular level, the disease is characterized by the accumulation of quasi-monoclonal B cells in the blood and lymphoid tissues. Despite its standard molecular phenotype, the disease represents heterogeneous clinical outcomes—some patients exhibit aggressive disease course, requiring early treatment just after diagnosis, and succumb to death within few years. On the other hand, some patients experience a rather indolent disease type that requires minimal or no treatment and usually survive a normal lifespan [1,2,3,4]. Consistent with this heterogeneity, no single causative agent has been identified for CLL. Rather, several genetic and immunogenetic modifications and chromosomal abnormalities have been associated with the disease that, either singly or in combination, predispose an individual to the disease.

Initially, CLL was believed to be a disease originated from immature B cells with minimal self-renewal capacity—the accumulation resulting from a defect in apoptotic pathways [5,6,7]. An increasing body of evidence changed this notion. First, the surface expression profile of CLL B cells shows that they are CD19^+^, CD5^+^, CD23^+^, IgM^+^, IgD^low^, and CD79b^low^-, indicating a mature B-cell phenotype [8]. Second, the presence of somatic hypermutation (SHM) in the immunoglobulin (Ig) gene in certain CLL cases suggests that the cells are antigen-experienced and not immature [9,10]. Third and most importantly, the tremendous clinical efficacy of B-cell receptor (BCR) signaling inhibitors, such as Ibrutinib (inhibitor of Bruton’s tyrosine kinase, BTK) or Idelalisib (inhibitor of phosphoinositide 3-kinase, PI3K), in reverting CLL pathology proves the immense importance of BCR-derived survival signals for the sustainability of the CLL cells (reviewed in [1,2,3,4,11,12]). Hence, the clonal accumulation is not attributed to failure in apoptosis, but rather it is a positive selection of the clones having an increased proliferation or self-renewing ability facilitated by receptor-derived stimulatory signals.

### 1.2. Classification of CLL—Factors Affecting the CLL Prognosis

Amidst the heterogeneity of the disease, some structurality emerged when it was found that CLL cases could be divided into two distinct categories, depending on the presence of SHM in the variable (V) region of the Ig heavy-chain (*IGHV*) gene of the B-cell clones [13,14,15]. Cases which show >98% identity to the germ line Ig sequence are termed as unmutated CLL (U-CLL), while those having <98% identity are termed as mutated CLL (M-CLL). Importantly, this classification represents robust prognostic value, with U-CLL cases generally being much more aggressive in clinical course compared with M-CLL, which is more indolent in nature [16,17]. This classification, however, does not cover the entire diversity of clinical outcomes. Some M-CLL cases were found to be more aggressive than expected. An inspection of their Ig sequence revealed that they carried borderline mutational load and therefore were intermediate between U-CLL and M-CLL [18].

In addition to *IGHV*’s mutational status, the expression of certain proteins was found to be of prognostic relevance. The transmembrane glycoprotein CD38 involved in the synthesis and hydrolysis of cyclic ADP-ribose was shown to be upregulated in U-CLL compared to M-CLL and was associated with poor outcome [16]. However, further investigation identified conflicting results, as some studies indicated no variation in CD38 expression among U-CLL and M-CLL [19] while others demonstrated that the expression of the protein varied during the course of the disease [20,21]. This conflict may arise due to methodological differences, including sample preservation techniques. Another important biomarker is the T-cell receptor zeta chain-associated protein (ZAP70), which is rarely expressed in normal B cells but is expressed in CLL B cells [7,22]. Gene expression analysis has shown that *ZAP70* is upregulated in U-CLL as compared to M-CLL. Recently, the expression of integrin α4 (CD49d), the α chain of the integrin heterodimer CD49d/CD29, is also found to be associated with CLL progression [23,24,25]. CD49d is involved in the homing of CLL cells to lymphoid niches and provides pro-survival signals for the survival and proliferation of the malignant cells. In certain CLL cases, CD49d shows a bimodal expression pattern, having both CD49d^+^ and CD49d^−^ CLL B-cell populations in the same sample, with the CD49d^+^ population having more proliferative capacity [26]. Additionally, the CD49d^+^ population in this bimodal sample increased with time, indicating that it could be a marker for the progression of the disease [26]. 

Several cytogenetic aberrations are also observed in CLL. The most frequent is the deletion of the long arm of the chromosome 13’s band 14(del 13q14), which contains two microRNAs, miR15a and miR16a (reviewed in [27,28]). Isolated del 13q is usually associated with good prognosis and low-risk CLL. The next most-abundant aberration is trisomy 12, which is associated with intermediate risk. Patients with trisomy 12 usually have additional aberrations, such as trisomy 19, or other deletions such as 11q, 13q, 14q, or 17p. The loss of chromosome 11’s long arm (del 11q) and chromosome 17’s short arm (del 17p) are the two most high-risk groups of CLL that show the minimal overall survival (OS) and treatment-free interval (TFI), owing to the loss of the DNA damage-repair gene Ataxia telangiectasia, mutated (*ATM*) in del11q, and the tumor-suppressor gene *TP53* in del 17p cases, respectively. Notably, the percentage of del 17p cases is lower at the initiation of the disease but goes considerably high in patients receiving chemotherapy, indicating that this aberration is related to chemorefractory CLL.

Recurrent somatic mutations in several cellular proteins are also observed in CLL. The single-pass transmembrane receptor protein NOTCH1 is frequently mutated in CLL (reviewed in [29]). The majority of mutations (~80%) are observed in the C-terminal proline (P), glutamic acid (E), serine (S), and threonine (T)-rich PEST domain that regulates the proteasomal degradation of the active NOTCH intracellular domain (NOTCH-ICD), leading to its prolonged half-life [30]. *NOTCH1* mutations generally represent poor prognostic CLL as they are frequently associated with *TP53* mutation, trisomy 12, and U-CLL (reviewed in [29]). The RNA-splicing factor SF3B1, which is an essential component of the U2 small-nuclear ribonucleoprotein particle (snRNP), is also frequently mutated in CLL and is associated with aggressive disease outcome (reviewed in [31]). The occurrence of *SF3B1* mutation in CLL patients is found to increase with the course of the disease, the maximum incidence being at the treatment phase or in chemorefractory CLL, indicating that *SF3B1* mutations may be acquired during therapy [32]. Interestingly, mutations in *NOTCH1* and *SF3B1* are also traced to multipotent hematopoietic stem/progenitor cells in some CLL cases, indicating that the acquisition of these mutations is the primary transformation event that most likely provides survival advantage for these transformed progenitors, which in turn acquire further disease-modifying mutations along the lymphoid differentiation and maturation process [33,34]. Although very rare, the translocation of the *MYC* proto-oncogene is also observed in CLL, mostly in association with the del 17p. *MYC* overexpression and *TP53* deletion in these so-called ‘Double-Hit’ CLL are extremely high-risk in nature (reviewed in [35]). 

In addition to the factors mentioned above, CLL cells demonstrate characteristic epigenetic modification, specifically DNA-methylation pattern [36]. In U-CLL, certain tumor-suppressor genes were found to be hypermethylated, while genes involved in cell proliferation and tumor progression were hypomethylated, as compared to M-CLL [36]. Analysis of the genome-wide methylation pattern in CLL cells as well as normal B-cell subsets identified close correspondence of M-CLL with memory B cells, while U-CLL closely matches naïve B cells, indicating their potential cellular origin [37]. Interestingly, some CLL cases, predominantly the M-CLL cases with relatively low mutational load, showed a DNA methylation pattern intermediate between the memory and naïve B cells, which suggests that they are probably derived from antigen-experienced extrafollicular B cells [37]. Importantly, these three epigenetic subgroups, namely, the memory-like (mCLL), naïve-like (nCLL), and intermediate (iCLL) CLL manifest distinct clinical outcomes [37,38]. The nCLL cases have much shorter time to first treatment (TTFT) and OS compared with the other two groups, indicating its aggressive, poor-prognostic nature. The mCLL cases, on the other hand, have the longest TTFT and OS and are rather indolent ones. The iCLL lies in between mCLL and nCLL. However, this subtype is not a mere mixing between mCLL and nCLL; rather, it has some specific characteristics, such as borderline mutational load and overrepresentation of the mutation of the splicing factor *SF3B1*, which suggests that this subclassification has its own clinical significance [38].

Thus, together with the mutational load of the *IGHV* gene, different cytogenetic aberrations as well as biomarker expressions and epigenetic classification can help define the course of the disease.

### 1.3. CLL Subsets

One of the major hallmarks of CLL is the finding that many of the unrelated CLL cases express almost identical Ig sequences, the so-called stereotyped BCR on their malignant B-cell surface (reviewed in [39]). Given the great diversity of the possible variable (V), diversity (D), and joining (J) gene recombination in heavy chain, along with the V–J recombination in light chain, B cells can produce ~10^12^ distinct Ig gene combinations. Thus, a similarity in the expressed BCR is expected to be extremely low. Considering this remote possibility, the expression of stereotyped BCR in CLL indicates that some common factors guide the evolution of these cells [40]. This could be exposure to some common antigens, self- or poly-reactive antigens, or interaction with the microenvironment, reinforcing the idea that CLL is a disease of activated B cells. Interestingly, not all CLL cases can be stratified into stereotypes. In fact, about one-third of all CLL cases show this property, while for the other two-thirds, such similarity in Ig gene sequences does not exist.

In the past 20 years, huge initiatives have been taken to categorize the sequence information of different CLL cases from different parts of the world. With the increasing ease of sequencing and the advent of automated bioinformatic tools to annotate them, it is now possible to classify the CLL cases (one-third of all CLL cases that belongs to stereotyped CLL) into 19 major subsets depending on the *IGHV-D-J* gene recombination pattern and the length and amino acid composition at the heavy-chain variable complementarity determining region 3 (HCDR3) (reviewed in [2,39,41]). Remarkably, the members of each subset share common disease features such as genetic mutations, chromosomal abnormalities, epigenetic condition, gene expression profiles, defects in certain signal transduction pathways, and most importantly, the clinical prognosis and outcome. For instance, subset 2, which is the largest CLL subset, is defined by the utilization of *IGHV3-21* and *IGHJ6* genes, a nine-amino acid-long VH CDR3 and the presence of a negatively charged amino acid (aspartic acid (D) or glutamic acid (E)) at position 3 of VH CDR3. This subset contains mostly M-CLL, having a very distinct pattern of SHM. Additionally, high incidence of del13q, del11q, mutations in the splicing factor *SF3B1*, and low incidence of TP53 dysfunction are characteristics for this subset. The very aggressive nature of this subset remained rather enigmatic until recently, when it was observed that a subgroup of *IGHV3-21*-expressing cells can trigger autonomous B-cell signaling that promotes better survival of these leukemic cells [42,43]. Contrary to this, subset 4 is characterized by the use of *IGHV4-34* and *IGKV2-30* genes and a long positively charged VH CDR3 that resembles pathogenic anti-DNA antibodies (reviewed in [39]). They are class-switched IgG-expressing clones and mostly belong to M-CLL, with a very limited number of genetic aberrations, and are consequently very indolent in nature. 

Very recently, an in-depth study of BCR stereotypy in a very large cohort of ~30,000 patients reported that 41% of all CLL cases can be classified as stereotyped subsets—29 of which are major subsets [44]. The stereotypy is mostly evident in U-CLL, with highly conserved HCDR3 within subsets, while M-CLL represents less stereotypy, with relatively degenerate sequence motifs in the HCDR3. Intriguingly, in this very large cohort, several minor subsets, the so-called satellites, are identified that are structurally linked to the major subsets and essentially have clinical outcomes similar to their corresponding major subsets. This concept of satellite subsets allows the inclusion of more sequences in the stereotyped classification, which in turn can provide better prediction in terms of clinical outcome for an increased number of CLL cases.

Thus, classification of CLL into different subsets depending on their BCR stereotypes not only provides a homogeneous profile to this otherwise heterogeneous disease but also help to understand the different pathological mechanisms that underlie each category and thus their clinical implications.

## 2. Importance of Light-Chain Sequence in CLL

The majority of CLL research is focused on the sequence of the Ig heavy chain. This includes the analysis of the mutational status of VH gene for U-CLL and M-CLL classification or the analysis of the length and amino acid composition of the VH CDR3 region for stereotyping. Increasing evidence points to the fact that light-chain sequence also affects the outcome of CLL. In 2003, Tobin et al. reported that the *IGHV3-21* gene, overrepresented in a cohort of M-CLL in Scandinavian population, shows remarkable pairing bias for *IGLV3-21* (formerly known as *IGLV2-14*) and is associated with poor outcomes despite being M-CLL [45]. Few years later, a light-chain profiling of 276 CLL patients revealed that, in addition to the heavy-chain genes, the usage of *IGK* (expressing κ-light chain) and *IGL* (expressing λ-light chain) gene repertoires are also restricted in CLL [46]. Of note, the biased usage of light-chain gene repertoire is found in normal B cells, particularly in fetal B cells. However, compared with normal healthy B-cell repertoire, the *IGKV1-8* and *IGKV2-30* genes were significantly overrepresented in the κ-expressing CLL cohort, while *IGLV3-21* and *IGLV2-8* were the two most frequently used *IGLV* gene in the λ-expressing CLL cohort. Additionally, skewing to longer CDR3 (more than 11 amino acids) was observed for both IGKV-J and IGLV-J rearrangements, specifically for those involving the *IGLV3-21* gene. This trend of longer LCDR3 (CDR3 of λ light chain) is not evident in *IGLV3-21* sequences in other diseases, such as multiple myeloma, reflecting that it is a CLL-specific event [46]. Shortly after, further evidence emerged supporting the biased use of *IGLV3-21* gene in *IGHV3-21*-expressing CLL cases in a larger cohort of patients encompassing different geographical regions, indicating that *this IGHV3-21/IGLV3-21* pairing is a general notion of CLL worldwide [47,48]. Interestingly, the majority of the *IGLV3-21*-expressing cases also exhibited Vκ rearrangements, which were non-productive either due to κ-deletion element (KDE) rearrangement or out-of-frame Vκ rearrangement, indicating that the cells actually followed the traditional light-chain rearrangement pathway in which autoreactive BCRs in immature B cells activate secondary light-chain gene rearrangements, thereby resulting in receptor editing and alteration of the autoreactive BCR specificity [48]. This also reflects the fact that *IGHV3-21/IGLV3-21* pairing is positively selected, most probably because it provides some survival benefit to the CLL cells as compared to other *IGHV3-21/IGKV* pairs. In fact, the median survival of patients expressing the *IGHV3-21/IGLV3-21* pair was lower compared with other M-CLL cases, irrespective of the mutational load of the *IGHV3-21* gene [47,48]. This clearly demonstrates that *IGHV3-21/IGLV3-21* positive cases do not fall within the traditional classification of U- and M-CLL.

Subsequently, it was shown that in CLL, the SHM pattern and load in the light chain differ significantly with the use of specific *IGK/IGL* gene and length of KCDR3/LCDR3 just as in the case of the heavy chain [49]. The most striking SHM pattern was observed for the subset 2 candidate *IGLV3-21* and subset 4 candidate *IGKV2-30* genes. *IGLV3-21*, in analogy to its partner *IGHV3-21*, was found to carry minimal/borderline mutations. However, a stereotyped SHM substituting S-to-G at LCDR3 was found in the majority of subset 2 *IGLV3-21*-expressing cases, compared with non-subset-associated CLL and non-CLL cases [49]. Similarly, a stereotyped site-specific insertion of aspartic acid residue at the κ framework region 3 (KFR3) was found for the majority of *IGKV2-30* genes in subset 4, which might be important for balancing the highly positively charged HCDR3 region of *IGHV4-34*-expressing subset 4 cases [49]. Taken together, these evidences clearly show that, similar to heavy-chain, stereotyped SHMs can also be found in light chains and may confer certain survival advantages for the malignant cells. Thus, light-chain gene sequences are equally important as the heavy-chain in order to understand the disease’s evolution, maintenance, and outcome.

Recently, the *IGLV3-21* gene has been interpreted as an independent prognostic marker for poor outcome CLL, irrespective of its pairing heavy-chain counterpart [50,51,52]. Using four different study cohorts, Stamatopoulos et al. observed that *IGLV3-21* gene is overrepresented (28%) in the high-risk cohorts compared to its overall incidence in CLL (7%) [50]. Interestingly, *IGLV3-21*-positive cases that belong to subset 2 demonstrate very aggressive disease course, with considerably shorter treatment-free survival (TFS) and OS similar to U-CLL cases, although the majority of them are M-CLL. On the contrary, the prognosis of non-subset 2 *IGLV3-21*-positive cases is dependent on the *IGHV* mutational status, as in other CLL cases. Additionally, we also found that the genes involved in translation processes and Myc target genes are upregulated in *IGLV3-21*-expressing cases but that they were largely independent of the *NOTCH1* mutation or *SF3B1* mutations. The molecular mechanism underlying this unusually poor prognostic property of *IGLV3-21* was elucidated in subsequent studies [43,51,52]. However, before discussing the molecular characteristics of *IGLV3-21*, the survival mechanisms of CLL B cells must be reviewed.

## 3. Factors Affecting CLL-Cell Survival

It is well accepted that in CLL, malignant B cells possess a survival advantage that leads to their accumulation. Several lines of evidence prove that BCR activation is essential for the development of the disease. The presence of SHM in the Ig gene sequences in M-CLL patients indicate that the cells are antigen-experienced post-GC cells. The restricted use of the Ig gene repertoire and the expression of stereotyped BCR with highly similar or sometimes identical sequences in otherwise-unrelated CLL cases also support the notion that some common factors/stimuli guide the proliferation, survival, and also the selection of these stereotyped B cells in the course of the disease. It is not clearly known which factors are responsible for such transformation. Studies have shown that many CLL BCRs, especially the U-CLL BCRs are poly-reactive in nature, as they recognize multiple antigens including self- or auto-antigens, such as cell debris or nucleic acids, lipids from dead cells, etc. Alternatively, some latent viruses or commensal bacteria can serve as antigens that repetitively stimulate B cells. Indeed, some CLL BCRs show structural similarity with antibodies that recognize the carbohydrate patterns of bacterial or viral coats [53]. No matter what is the source of the antigen, the recognition of these antigens by BCR leads to the activation of cascades of signaling events that finally triggers gene expression alteration through certain transcription factors and results in selective proliferation of the antigen responsive cells. Thus, antigenic stimulation potentiates the clonal evolution of the cells, which then perpetuates the development of the disease. The strongest evidence for the involvement of BCR signaling in CLL comes from the fact that BCR-signaling pathway inhibitors offer the most effective and successful treatment for CLL (reviewed in [1,2,3,4,11,12]). It should, however, be noted that testing the antigen specificity of Igs expressed on CLL B cells using recombinant in vitro techniques for the production of antibodies may not reflect the binding characteristics of the CLL BCR, which is membrane-associated and may therefore have a different antigen-binding capacity compared to the respective soluble antibody [54].

Apart from the BCR derived signals, several co-stimulatory signals, either in the form of direct cell–cell interaction or through soluble factors, are necessary for the survival of the CLL cells. They not only provide signal for proliferation but also help in the migration and homing of the leukemic cells to lymphoid niches where they can divide. Recent studies have shown that CLL cells that reside in the periphery are quiescent with surface phenotype CXCR4^high^CD5^low^, while those residing in the lymph nodes are actively proliferating (CXCR4^low^CD5^high^) [55,56]. Several co-stimulatory signals have been proposed to play a role in CLL (reviewed in [57,58]). For instance, stimulating CLL cells in vitro with CD40L and interleukin 21 (IL21) leads to their proliferation, indicating that a CD40-signaling pathway is probably involved in the disease [59,60]. The presence of CD40L-expressing CD4^+^ T cells in the lymphoid follicles bearing the CLL cells also indicate that CLL cells require CD40 co-stimulatory signal for their proliferation [61]. Another important signaling cascade is Toll-like receptor (TLR) signaling. TLRs are pattern-recognition receptors (PRR) belonging to the innate immune system and are activated upon encountering specific molecular patterns on microorganisms. Given the fact that CLL can be triggered by infection, the activation of TLR signaling on CLL B cells is not surprising [62]. Specifically, TLR9 expression is increased in CLL cells, compared with healthy B cells [63]. The recurrent mutations observed in the adaptor protein Myd88, which serves as a scaffold for TLR signaling, also supports the role of this pathway as a disease-modifying factor [64].

Besides antigen-mediated stimulation, BCR can also be activated in a novel antigen-independent process called autonomous B-cell signaling. In 2012, our group demonstrated for the first time that CLL B cells possess the unique capacity to activate BCR signaling, even in the absence of any ligands [42]. In this study, an inducible cell system derived from *Rag2^−/−^ λ5^−/−^ SLP65^−/−^* triple knock-out (TKO) mouse bone marrow was used [65]. The loss of *Rag2* and *λ5* genes prevented the expression of any BCR on the surface of these cells. However, when transduced with plasmids expressing pre-recombined heavy- and light-chain Ig sequences, the cells were able to ectopically express the respective BCR. Additionally, the lack of SLP65 prevented any BCR-derived signals from being transmitted. Therefore, the cells were maneuvered to express a tamoxifen-inducible SLP65 protein, making the whole system effectively controllable, to study BCR signaling under different conditions. Using this system, we demonstrated that BCRs derived from CLL patients, irrespective of U-CLL or M-CLL, can trigger receptor activation-induced calcium release without any external antigenic stimulation of the BCR [42]. This was, however, not observed for BCRs derived from other B-cell malignancies, such as multiple myeloma, mantle cell lymphoma, marginal zone lymphoma, or follicular lymphoma. Interestingly, in contrast to the previously observed fact that autonomous signaling is related to the poly-reactivity of the BCR [66], the current sets of CLL BCRs were found to be mostly non-polyreactive [42]. Detailed investigation of the sequences of these BCRs revealed a conserved epitope at the framework region 2 (FR2) of their VH domain that had sequence homology to a peptide sequence previously shown to bind CLL receptors [67]. Mutation in specific amino acids in this epitope, namely, Valine to Glycine at position 37 (V37G) and Arginine to Alanine at position 38 (R38A), led to the complete removal of autonomous signaling, while mutation at R43A outside of this epitope had no effect on autonomous signaling [42]. Notably, none of these mutations affected the conventional signaling of these BCRs induced by BCR ligation with anti-BCR antibodies, indicating that the internal epitope responsible for autonomous signaling is not required for surface expression or conventional function of the BCR. This is further supported by the observation that V37G mutation in a non-autonomously active receptor did not affect its ligand binding and subsequent signaling. Thus, it was concluded that a BCR-intrinsic motif on CLL B cells is able to bind and activate neighboring BCRs in the absence of any external antigens and provide critical support for the survival and proliferation of the malignant clones. In line with this, primary CLL B cells showed an elevated basal–calcium signaling, compared with healthy normal B cells [42]. Taken together, this cell-autonomous signaling marks a new paradigm for the CLL cell-survival mechanism.

## 4. Structural Basis of Autonomous Signaling

The idea that CLL B cells are able to stimulate their growth by ligand-independent BCR signaling opens up new paths for CLL research. The massive heterogeneity of the disease cannot be explained by a single unifying mechanism. Therefore, this raises the question whether autonomous signaling is equally involved in different subsets of CLL, and if so, then what makes the differences in the outcome. The answer to these questions came in 2017, when Degano and colleagues provided a structural basis for autonomous signaling and demonstrated that homotypic BCR–BCR interaction in CLL has distinct subset-specific features [43]. The study undertook crystallographic analysis of the Fab (Fragment antigen binding) fragments of CLL BCRs belonging to the indolent subset 4 and aggressive subset 2 CLL. In subset 4, the interaction took place through the HCDR3 of the ‘receptor’ (paratope) Fab and the heavy-chain framework region 1 (V_H_FR1) and C_H_1 of the ‘antigen’ (epitope) Fab. The interactions were stabilized by specific salt bridges, hydrogen bonds, and van der Waals’ contacts. Interestingly, the recurrent SHMs observed in subset 4 were found to be important for maintaining these stabilizing interactions. For example, the characteristic SHMs Y31H and Q43H observed in the *IGKV2-30* light-chain gene in subset 4 cases help stabilizing the active-site architecture, while the G31E mutation in the heavy chain enhanced the interaction surface with the epitope, which was otherwise not possible by the germline-encoded residues. Moreover, the structural analysis also found the justification of class-switching to IgG as observed in subset 4 CLL cases. The positively charged K214 residue in the C_H_1 of the BCR formed hydrogen bonds with the two tyrosine residues, Y110 and Y111, of the light chain and stabilized the BCR–BCR self-association. This K214 residue was not conserved either in human IgM or IgE or in mouse IgG constant domain. Consequently, expressing the subset 4 BCR as mouse IgG abolished autonomous signaling [43].

The basis for autonomous signaling in subset 2 was found to be different from that of subset 4. Unlike subset 4, the homotypic BCR–BCR interaction in subset 2 was mediated by the IGLV3-21 light chain, specifically through the LCDR1-LCDR2 loop of the ‘receptor’ BCR and a composite region spanning LFR1 and the linker region between VL–CL of the ‘antigen’ BCR [43]. A key finding in the crystallographic analysis of subset 2 was the identification of an acquired mutation in the light chain. This so-called R110 mutation resulted from a G to C substitution at the last nucleotide of the V–J segment and led to an amino acid change from glycine to arginine [43]. Specific hydrogen bonding between this R110 at the linker region in the epitope with D50 at the LCDR2 in the paratope and a salt bridge established between K16 of the LFR1 in the epitope and D52 at the LCDR2 in the paratope stabilized the BCR–BCR interaction. Mutation of these residues mitigated autonomous signaling, as evidenced by the cessation of intracellular calcium release in the inducible TKO cell system, indicating their importance for the interaction. Notably, IGLV3-21 is the only light chain that contains the “YDSD” motif with two D residues (D50 and D52) in the LCDR2 that are required for autonomous signaling. Therefore, this explains why the *IGLV3-21* gene is preferentially used in subset 2 cases. Similarly, the finding that R110 at the junction of the VL and CL region is centrally involved in the mutual BCR–BCR interaction explains the acquisition of the recurrent R110G mutation as observed in subset 2 cases, as the germline-encoded G110 cannot provide the necessary interaction to support autonomous signaling.

The structural study also provides insights into the different clinical outcomes of the two subsets [43]. The relatively lower number of stabilizing molecular interactions and high equilibrium-dissociation constant (K_d_) in subset 2 indicate that the homotypic BCR–BCR interaction is weak and transient in this case. On the contrary, subset 4 is characterized by strong and persistent interaction with a higher number of stabilizing molecular interactions and lower K_d_ value. This strong and continuous BCR stimulation might desensitize the CLL cells and render them in an anergic state where they become mostly functionally nonresponsive. This leads to an indolent nature of the disease, as seen in subset 4. On the other hand, the weak and short-lived interactions in subset 2 provide periodic stimulation pulses leading to the steady proliferation of the CLL cells. This in turn results in an aggressive disease course, as observed in the case of subset 2 CLL. Importantly, the presence of additional disease-modifying mechanisms, such as genetic mutations or cytogenetic aberrations, can modulate the effect of autosomal signaling, leading to a heterogeneous clinical outcome, even within the same subset of the disease.

Thus, autonomous signaling seems to be a general phenomenon of CLL, and it is associated with essential structural requirements. Interestingly, the cells with autonomously active BCRs can impart different effects, depending on the strength of the homotypic interaction. In addition, it is important to mention that autonomous BCR signaling provides the very essential survival signal for CLL cells independent of any ligand for the BCR. This, however, does not necessarily imply that the cells are independent of tumor microenvironment-dependent cues. The complex interaction between the CLL cells with the tumor microenvironment still remains very crucial for their survival.

## 5. IGLV3-21^R110G^ Defines a New Category of CLL

Taken together, the crystallographic study pointed out the importance of the critical R110G mutation in the homotypic interaction in subset 2 CLL [43]. Soon it was realized that patients carrying IGLV3-21^R110G^-mutated immunoglobulin gene manifest a highly aggressive disease course similar to the U-CLL [51,52]. Using both DNA sequencing and a novel antibody-based screening approach, our group showed for the first time that IGLV3-21^R110G^-expressing CLL cases have significantly lower OS, TTFS, and progression-free survival (PFS) and thus mostly follow the course of U-CLL, compared with IGLV3-21^R110G^ negative M-CLL cases [51]. Interestingly, IGLV3-21^R110G^-positive cases belong to both U-CLL and M-CLL categories, but their prognosis was independent of an *IGHV* mutational status as the clinical parameters for IGLV3-21^R110G^-positive M-CLL and IGLV3-21^R110G^-positive U-CLL were virtually identical, indicating that IGLV3-21^R110G^ is a standalone prognostic marker [68]. Indeed, many of the IGLV3-21^R110G^-positive cases actually possessed the favorable del13q genetic aberration, while the unfavorable del17p or del11q was rarely seen [51]. Our observations were further supported by other studies that also reported adverse disease progression in IGLV3-21^R110G^-expressing cases irrespective of their mutational status [52,69], even in familial CLL cases [69]. In one of these studies, the epigenetic landscape, specifically the DNA methylation pattern of the IGLV3-21^R110G^-expressing CLL cells, was analyzed in comparison with the non-IGLV3-21^R110G^ CLL cells [52]. It was found that, compared to the naïve (nCLL) or the memory (mCLL) CLL categories, the intermediate CLL (iCLL) category was over-represented in IGLV3-21^R110G^-positive CLL. The clinical features of iCLL as a whole were intermediate between nCLL (aggressive, similar to U-CLL) and mCLL (indolent, similar to M-CLL). However, when separated on the basis of IGLV3-21^R110G^ mutation, the IGLV3-21^R110G^-positive iCLL cases showed TTFT and OS similar to aggressive nCLL, while the non-IGLV3-21^R110G^ iCLL cases coincided with the indolent mCLL. In other words, IGLV3-21^R110G^ splits iCLL into two distinct prognostic categories, emphasizing its own independent clinical value [52]. Importantly, both studies found that IGLV3-21^R110G^-positive cases were not restricted to subset 2 as the majority of them belonged to non-stereotyped CLL and hence remain unclassified as of now [51,52]. Therefore, IGLV3-21^R110G^ emerges as a new category of CLL classification that otherwise cannot fit into the conventional classification and stereotyping of the disease. 

To understand the reason behind the highly aggressive nature of IGLV3-21^R110G^-expressing cases, we profiled the surface-marker expressions of IGLV3-21^R110G^ cases, along with IGLV3-21^R110G^-negative U-CLL and M-CLL cases [51]. As expected from their clinical courses, the IGLV3-21^R110G^-expressing cells closely matched the surface-expression profile of the U-CLL. In addition, they possess distinct characteristics with increased CD23 and CD43 and reduced CD22 expressions. In line with this, Nadeu and colleagues also found that the gene expression profile of IGLV3-21^R110G^-positive cells is similar to that of nCLL, with a characteristic up-regulation of WNT5A/5B [52]. Elevated WNT signaling might cause enhanced chemotaxis and better proliferation of the CLL cells, leading to aggressive disease course [70,71]. In addition, a higher prevalence of SF3B1 mutation and ATM mutation was also observed in both the studies [51,52]. Since the R110G mutation was first described in the context of autonomous signaling in subset 2 CLL, we tested to see if IGLV3-21^R110G^ cases, which do not belong to subset 2, manifest ligand-independent autonomous activation of BCR. Expectedly, we found that IGLV3-21^R110G^-expressing BCRs were autonomously active, while the unmutated IGLV3-21^G110^ did not possess this capacity [51]. 

An important question in this context is how this R110G point mutation is acquired. Structurally, the mutation is located at the junction of the VL–CL region of the IGLV3-21 gene [43]. Intriguingly, in our study, we found that the expression of activation-induced cytidine deaminase (*AICDA* or *AID*), the enzyme required for the SHM, was up-regulated in IGLV3-21^R110G^-expressing cells similar to what is observed for M-CLL [51] (Figure 1). This indicates that R110G might have generated through ongoing SHM. AID deaminates cytosine (C) residues within the motif WRC (W = A/T, R = A/G) or WGCW, the latter being an overlapping motif for both DNA strands [72]. The major sites for AID-mediated deamination, the so-called AID hotspots, are generally found in the CDR regions of the V_H_ or V_L_ genes that are targets for SHM in normal B cells. Such hotspots are not observed in the junctional regions, indicating that R110G is outside of the AID hotspot. However, this does not exclude the possibility of AID-mediated mutation. In fact, in CLL, an inverse correlation is observed between the mutational frequency and the number of overlapping AID hotspots in the context of heavy-chain variable genes [73]. An alternative possibility could be a defective end joining during the double-strand break (DSB) repair occurring in VL–CL joining, specifically in an error-permissive environment where proteins involved in DSB repair are mutated [74] (Figure 1). Prevalent genomic mutations in the essential DNA damage-repair proteins such as ATM and/or SF3B1, as observed in IGLV3-21^R110G^ cases, predispose the developing early B-cell progenitors to faulty VL–CL joining, thereby acquiring the R110G mutation and developing highly aggressive CLL. Since the G to C mutation at the end of the VJ segment alters the consensus sequence required for efficient splicing of the VJ exon to the CL exon of the light chain, it is tempting to speculate that the mutated SF3B1 is somehow involved in the proper splicing of the mutated VJ segment. Intriguingly, in certain IGLV3-21^R110G^-positive cases, apart from the dominant R110G expressing clone, several bystander clones are also detected that contain the R110G mutations but differ in IGLJ gene usage [74]. This indicates that R110G mutation in both the clones evolved in parallel, probably through some common defective mechanism such as an error-prone VL–CL joining [74]. Alternatively, such differential J-gene usage may also result from the secondary light-chain gene rearrangement or receptor editing in an attempt to modulate the reactivity of the original BCR. However, it is also important to point out that cases where IGLV3-21^R110G^ is the sole mutation without any other driver mutations or genomic aberrations are known, for which the clinical outcome is equally aggressive as that of the other IGLV3-21^R110G^ cases with additional mutations [51,52]. The mechanism of acquiring the sole R110G mutation in these cases is not understood yet.

The R110G mutation was first described in the context of subset 2 CLL, where *IGLV3-21* predominantly pairs with *IGHV3-21* [43]. However, majority of IGLV3-21^R110G^-positive cases are found to be non-subset 2, having other IGHV partners [51,52]. For instance, the minor subset 169, which is currently considered as a satellite of subset 2, contains an *IGHV3-48/IGLV3-21* pair with very similar VH CDR3 sequence with subset 2 and similar aggressive clinical outcome [44,75]. Interestingly, apart from the acquired IGLV3-21^R110G^ light-chain mutation, the *IGHV3-48* also possesses crucial SHMs, namely, D31S and D53S, that are required for cell-autonomous signaling [76]. Mutating these two D residues back to their germline S residues abolishes autonomous signaling, even in the presence of the R110G mutation in the light chain. This highlights the fact that, depending on the heavy-chain partner, R110G mutation may require additional mutations to sustain ligand-independent BCR signaling and concomitant survival advantage for the leukemic cells. Further studies are required to validate the relevance of these new residues as susceptibility factors and to elucidate the underlying mechanism for this observation. 

## 6. IGLV3-21*01/04 Are Potential Risk Alleles for Developing Aggressive CLL

One remarkable finding that emerged from the studies mentioned above is the fact that among the four different alleles of *IGLV3-21* gene, almost all IGLV3-21^R110G^ cases carry the *IGLV3-21*01* [51] or *IGLV3-21*04* allele [52]. Notably, the *IGLV3-21*04* allele, which differs from the *IGLV3-21*01* allele in only one nucleotide (Figure 2), is newly added to the ImMunoGeneTics (IMGT) database (release 202018-4) and was not annotated at the time of our publication [51]. The nucleotide difference (GAC in the *IGLV3-21*01* allele coding for aspartic acid and GAT in the *IGLV3-21*04* allele also coding for aspartic acid in the CDR3 region) does not alter the amino acid sequence of the encoded protein (Figure 2). Sequence analysis suggests that *IGLV3-21*01/04* are the two alleles that contain the essential K16 residue at the LFR1 region and the YDSD motif at the LCDR2 that are required for the homotypic BCR–BCR interaction, leading to autonomous signaling [51,52]. Indeed, changing the *IGLV3-21*01* allele for IGLV3-21^R110G^ cases to *IGLV3-21*02* containing a Q16 and *IGLV3-21*03* containing a DDSD motif diminished cell-autonomous signaling, indicating that only the *IGLV3-21*01* allele has structural privilege over the others [51]. The presence of another D residue at position 49 in the DDSD motif in the *IGLV3-21*03* allele probably disorients the D50 residue, making it unsuitable for homotypic interaction. However, some variations in the consensus YDSD motif seem to be allowed, as patients with IGLV3-21^R110G^ carrying the motif YDTD or FDSD also show a similar gene expression profile, epigenetic landscape, and aggressive disease course as that of the patients with IGLV3-21^R110G^ carrying YDSD motif [52]. The substitution of serine to structurally similar threonine at position 51 or the substitution of aromatic tyrosine residue to another aromatic phenylalanine at position 49 might not severely hamper the homotypic interaction. Thus, *IGLV3-21*01/04* alleles, upon acquiring only one mutation, namely R110G, can induce ligand-independent signaling, thereby stimulating cell proliferation and giving rise to aggressive CLL disease course. The other two alleles, *IGLV3-21*02/03*, require additional mutations apart from the crucial R110G to fulfil the structural criteria and are therefore not found in the cohort of IGLV3-21^R110G^ cases. Of note, all CLL cases carrying IGLV3-21 but not the R110G mutation (i.e., carrying either G110 or S110) were found to carry the *IGLV3-21*02/03* alleles, suggesting that molecular structures other than R110 might mediate BCR–BCR interactions in these cases [51].

Interestingly, R110G mutation is also prevalent in the normal healthy population [51]. However, in healthy donors, R110 was found to be associated with all light-chain variants except for *IGLV3-21* [51]. Among those few cells that do express IGLV3-21^R110^, they carry the *IGLV3-21*02/03* alleles and never the *IGLV3-21*01* allele [51]. Thus, in the healthy population, the presence of R110 cannot induce cell-autonomous signaling due to lack of structural support, which requires the *IGLV3-21*01* allele. Indeed, the striking underrepresentation of the IGLV3-21^R110^ in the normal population indicates that this combination is experiencing negative selection pressure in healthy B cells. On the other hand, in the CLL microenvironment, when a cell carrying the intrinsically autonomous signaling-prone *IGLV3-21*01/04* allele acquires the crucial R110 mutation, it experiences a survival advantage through autonomous BCR activation. Under the permissive growth conditions facilitated by different co-stimulatory signaling pathways and through loss/mutation of important cellular checkpoints, the clone rapidly evolves, sustains, and propagates, leading to an aggressive disease manifestation. Therefore, the alleles *IGLV3-21*01/04* can be considered as risk factors for the development of highly aggressive, poor prognostic CLL.

## 7. Concluding Remarks

The past decades have witnessed several landmark discoveries in CLL research. The remarkable correlation of the *IGHV* mutational status with the clinical course of the disease, the stereotypic classifications of the CLL BCRs and their prognostic implications, the discovery of cell-autonomous signaling as a survival mechanism for CLL cells—all these observations not only enriched our understanding of the disease’s pathology but also improved disease management. In this review, we discussed the relevance of Ig gene mutations as a potential risk factor in CLL in light of the recent finding of a single-point mutation in the Ig light chain. We discussed how acquisition of this R110G mutation on the specific light-chain alleles *IGLV3-21*01/04* renders them prone to autonomous signaling that ultimately explains the severe disease outcome observed in IGLV3-21^R110G^-expressing CLL cases. Until now, the stereotypic classification of CLL was based on the empirical sequence characteristics of the Ig genes. IGLV3-21^R110G^ is the first subgroup of CLL that is based on functional immunopathology. Therefore, redefining CLL subsets through inclusion of IGLV3-21^R110G^ as a separate subset will improve the prognostic power of the stereotyping. We also addressed the potential mechanisms by which R110G mutation may be acquired. However, further investigations are required to clearly delineate these mechanisms, which will provide new insights of the clonal evolution of the disease.

## Figures and Tables

**Figure 1 cancers-14-03045-f001:**
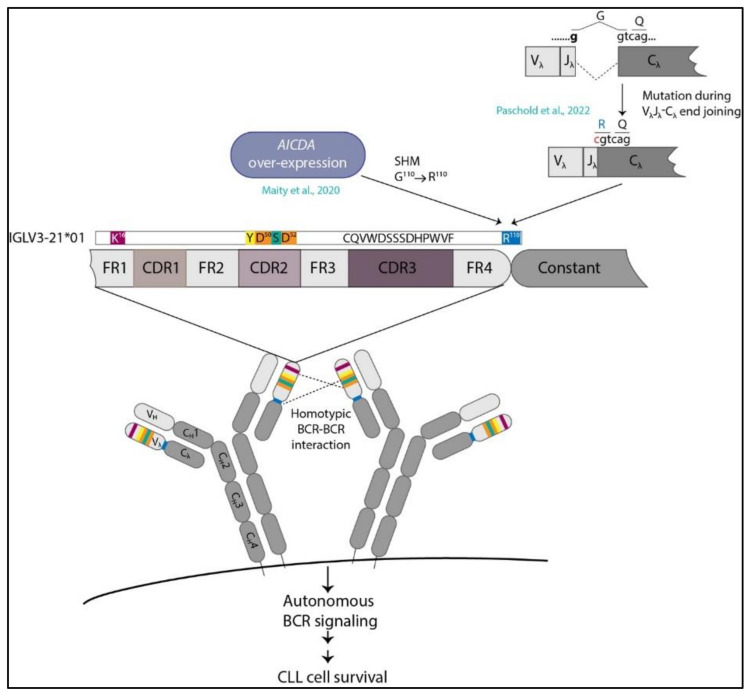
**Acquisition of R110G mutation on *IGLV3-21*01/04* alleles leads to aggressive CLL.** R110G mutation may be acquired through SHM induced by AID overexpression [51] or through defective VL–CL end joining during light-chain gene recombination [74]. The *IGLV3-21*01* (depicted in figure) and *IGLV3-21*04* alleles contain the K16 residue in the FR1 and two D residues at position 50 and 52 in the CDR2, which, along with the R110 at the VL–CL junction, fulfill all the structural requirements for homotypic BCR–BCR interactions. This in turn induces ligand-independent autonomous BCR activation, leading to proliferation and survival of the malignant cells that ultimately results in severe disease course.

**Figure 2 cancers-14-03045-f002:**
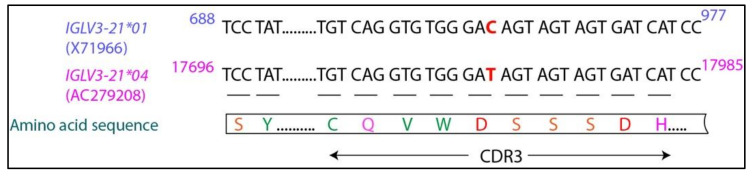
**Sequence alignment of *IGLV3-21*01* and *IGLV3-21*04* alleles.** The sequences were obtained from the IGMT database. The single nucleotide difference is marked in red. The encoded amino acid sequence as shown below the nucleotide sequences is identical for both the alleles.

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
