# Peer review of "Immunoglobulin Gene Sequence as an Inherited and Acquired Risk Factor for Chronic Lymphocytic Leukemia"

_cancers, 2022, doi:10.3390/cancers14133045_

Round 1
Reviewer 1 Report
Review of Datta and Jumaa
With great pleasure I read this comprehensive and informative review, which deals with many more aspects of CLL than the title suggests (I would change it to “…as inherited and acquired risk factor…”). It is written by the foremost experts in the field and will be a very valuable source of information for CLL researchers and scientists as well as clinicians interested in the biology of B cells and their transformation.
As a lot of the review deals with various aspects of CLL, I would suggest to put the “BCR-centric” statements also a little bit into the larger context of the disease. For example, in line 325 the authors write: “The massive heterogeneity of the disease cannot be explained by a single unifying mechanism.” As outline by them further below and above, in addition to BCR signaling CLL is highly heterogenous with respect to the genetic and epigenetic alterations, cell of origin etc, even if the autonomous BCR signals were highly similar. In this context it might be worth to point out studies suggesting that mutations at the level of the hematopoietic stem and progenitor cells play a role in CLL (Damm et al., Kikushige et al.). Furthermore, the strong focus on autonomous BCR signals creates to me the (surely unintended) impression that CLL and/or their precursors become self-sufficient. It would in my opinion therefore be useful to point out in the introduction that CLL cells remain exquisitely dependent on a complex tumor microenvironment, which they induce.
Minor comments:
- Line 114: “antigen experienced extra germinal center (GC) B cells”
Please define what is meant with extra germinal center B cells.
- Line 133: “Thus, a similarity in the expressed BCR is expected to be less than 1 in 1012 cases and therefore extremely unlikely”
I am not sure I can follow the mathematics of this argument. If we can make 1012 different BCRs, how can one derive and argument regarding similarity (how many BCRs are similar?) with this precision?
- Line 123-125: “Taken together, these evidences clearly show that even a single amino acid change in the light chain can alter the antigen binding specificity of the BCR and may confer certain survival advantages for the malignant cells.”
From the text I do not understand how the evidence clearly shows that the antigen binding specificity is changed.
- Line 449 to 451: “Prevalent genomic mutations in the essential DNA 449 damage repair proteins such as ATM and/or SF3B1, as observed in IGLV3-21R110G cases predispose the developing early B cell progenitors to faulty VL-CL joining thereby acquiring the R110G mutation and develop highly aggressive CLL.”
If the ATM and SF3B1 mutations would precede the B cell precursor stages (see Damm et al., Kikushige et al.) and the acquisition of the R110G mutation one would expect these then to be clonal. However, as the authors pointed out earlier, they are mutations often/mostly acquired at later stages of the disease. Are SF3B1 and ATM mutations (more often) clonal in R110G mutant CLL?
Reviewer 2 Report
This review article gives a very comprehensive and up-to-date summary regarding the alterations in immunoglobulin gene sequence along with an additional focus on IGLV3-21 in CLL. This scientific manuscript is solid evidence based, very well written, and easy to understand. It is an excellent review article. I do not have major comments and only have a couple of very minor comments.
Minor comments:
1. Please italicize the gene names, such as ATM, TP53 in line 85 and other places.
2. There may be some extra spaces (formatting issue) found in line 291 (after [40]), line 446 (after [71]), line 462 (after BCR), and line 465 (after [49,50]).
